# Molecular Pathways of Rosmarinic Acid Anticancer Activity in Triple-Negative Breast Cancer Cells: A Literature Review

**DOI:** 10.3390/nu16010002

**Published:** 2023-12-19

**Authors:** Evangelia K. Konstantinou, Athanasios A. Panagiotopoulos, Konstantina Argyri, George I. Panoutsopoulos, Maria Dimitriou, Aristea Gioxari

**Affiliations:** Department of Nutritional Science and Dietetics, School of Health Sciences, University of the Peloponnese, Antikalamos, 24100 Kalamata, Greece; e.konstantinou@go.uop.gr (E.K.K.); ath.panagiotopoulos@go.uop.gr (A.A.P.); kargyri@uop.gr (K.A.); gpanouts@uop.gr (G.I.P.)

**Keywords:** rosmarinic acid, breast cancer, natural products, polyphenols, antioxidants

## Abstract

Breast cancer is the most frequent type of cancer in women. Oncogenic transcription factors promote the overproduction of cellular adhesion molecules and inflammatory cytokines during cancer development. Cancer cells exhibit significant upregulation of antiapoptotic proteins, resulting in increased cell survival, tumor growth, and metastasis. Research on the cell cycle-mediated apoptosis pathway for drug discovery and therapy has shown promising results. In fact, dietary phytoconstituents have been extensively researched for anticancer activity, providing indirect protection by activating endogenous defense systems. The role of polyphenols in key cancer signaling pathways could shed light on the underlying mechanisms of action. For instance, Rosmarinic Acid, a polyphenol constituent of many culinary herbs, has shown potent chemoprotective properties. In this review, we present recent progress in the investigation of natural products as potent anticancer agents, with a focus on the effect of Rosmarinic Acid on triple-negative BC cell lines resistant to hormone therapy. We highlight a variety of integrated chemical biology approaches aimed at utilizing relevant mechanisms of action that could lead to significant clinical advances in BC treatment.

## 1. Introduction

Breast cancer (BC) develops from breast tissue when breast cells begin to grow abnormally [1]. After skin cancer, it is the most common cancer diagnosed in women, although it can occur in both men and women [1,2,3,4,5,6,7]. Usually, BC arises from the epithelial cells of the mammary gland and can metastasize through the breast to the lymph nodes or other parts of the body [8]. Clinically, BC can be divided into distinct subtypes based on hormone receptor status: Hormone receptor-positive (HR+), human epidermal growth factor receptor 2-positive (HER-2+), and triple-negative breast cancers (TNBC) [9,10]. This classification, based on hormone response, has prognostic implications and is a crucial factor in deciding between hormone and targeted therapy [9].

BC cells usually form a tumor, which can be detected as a new lump or mass that might cause pain, breast shape alterations, or fluid excreted from the nipple [11,12,13]. Other possible signs of BC are dimpling, peeling, or redness in the area of skin surrounding the nipple [1,2,3,4,5,6,7]. Many documented risk factors for BC have been identified, including age, gene mutations, family history, mammographic breast density, menstrual and menopausal history, radiation exposure, and lifestyle [14,15].

The diagnosis in its early stages involves measuring the expression levels of certain proteins in biopsy specimens, such as estrogen, progesterone, and HER2 receptors [16]. In addition, BC can be found using X-ray methods at a later stage, when the tumor formed is significantly larger [16]. Data from clinical trials suggest that radiation therapy and chemotherapy for BC remain the most effective methods of treatment [17]. The induced destruction of tissue and cell tumors following physical or chemical irreversible damage to the genetic material of cancer cells is a central goal of these therapeutic approaches [18]. However, the effectiveness of such treatments does not exceed 30%; in several cases, cancer cells show resistance after prolonged therapy [15,19]. As a result, in recent decades, research has concentrated on the development of medications that may target cancer cells while avoiding the molecular pathways that contribute to chemotherapeutic and radiation resistance [20].

During cancer development, various oncogenic transcription factors lead to the overproduction of cellular adhesion molecules and inflammatory cytokines [21,22,23,24]. Moreover, cancer cells demonstrate significant upregulation of antiapoptotic proteins (e.g., Bcl-2, Bcl-xL, etc.) or reduction in the expression of proapoptotic proteins (e.g., the p53 pathway), resulting in improved cell survival, enhanced tumor growth, and metastasis [25,26]. Chemoresistance is also enhanced by elevated expression levels of inhibitors of apoptosis (IAPs), a family of proteins significant in programmed cell death (PCD) [19]. Apoptosis alteration is responsible not only for tumor development and progression but also for tumor resistance to therapies [26]. The link between anticancer effects and the induction of cell death is a strongly investigated issue [27]. Targeting the cell cycle-mediated apoptosis pathway could be used for the development of innovative approaches to drug discovery and therapy [1,28].

The impact of natural products of plant origin, including polyphenols, on chemoprevention has been extensively studied, and it has been employed as an important approach in the fight against BC, offering indirect protection by activating endogenous defense systems [25,29]. In addition, the investigation of polyphenols in critical signaling pathways may shed light on the mechanisms of action of a polyphenol-rich diet [30]. Nuclear factor-kappa B (NF-κB) activation and activator protein-1 (AP-1) DNA binding are some of the processes sensitive to polyphenols’ activity [31]. Other changes to cellular signaling identified for the chemoprevention effect of polyphenols include estrogenic/antiestrogenic activity, antiproliferation, induction of cell cycle arrest or apoptosis, prevention of oxidation, and anti-inflammatory activity [32].

In this review, we present recent progress in the exploration of natural products as potent anticancer agents, focusing on the effect of a well-known polyphenol, Rosmarinic Acid, on MDA-MB-231 and MDA-MB-468 cells. These cells constitute TNBC, an aggressive and metastatic subtype unaffected by hormone therapy [33]. We present a range of integrated chemical biology techniques aimed at using relevant mechanisms of action that might lead to major therapeutic advancements in the treatment of BC [3,4,34,35,36,37,38,39,40,41,42].

## 2. The Role of Natural Products in Human Health

Natural products comprise a large group of chemical compounds produced by living organisms, including plants and foods, and are essential for survival or defense reasons [43,44,45,46,47,48]. Natural products can also be prepared through chemical synthesis and have been extensively used in the pharmaceutical and food industries [45]. With regard to natural products of plant origin, these are the basic ingredients of horticultural fruit and vegetable crops, wine, tea, extra virgin olive oil, medicinal plants (e.g., mint, rosemary, thyme, sage, and oregano), and beverages [45]. They contribute to plants’ health as growth agents and have a major role in defense mechanisms against infection and injury [49]. Furthermore, natural products are significant factors in pollination because of their colors [50].

Polyphenols are a major class of natural products that enhance food quality, offering bitterness, astringency, color, flavor, odor, and oxidative stability [32]. Polyphenols are secondary metabolites of plants, forming a large number of different substances with aromatic rings and hydroxyl groups [31]. They are mostly derivatives and/or isomers of flavonoids (flavones, isoflavones, flavonols, catechins), phenolic acids (hydroxybenzoic acids and hydroxycinnamic acids), stilbenes, and lignans [31,49]. They are reactive species toward oxidation, hence their description as antioxidants in vitro [51]. Polyphenols display a great variety of chemical behaviors due to their structural diversity, which is directly related to their abundance of properties [30]. Polyphenols are important constituents of the human diet and have attracted scientific attention as they display a wide variety of therapeutic properties, like antioxidant and free-radical scavenging activity, which lower the process of atherosclerosis and the risk of developing cardiovascular diseases [52,53]. Polyphenols contribute to redox homeostasis, a critical method for cells to deal with an excess of free radicals created by oxidative stress, which in turn is linked to aging and various degenerative illnesses, such as atherosclerosis, cardiovascular disease, type II diabetes, and cancer [31,54]. However, the exact mechanisms and the molecular targets of their function are not clear in many cases [30].

Natural products and polyphenols found in foods are currently being developed for health benefits and are described by several terms, such as nutraceuticals, functional foods, dietary supplements, and phytochemicals [14]. The health effects of polyphenols depend on the amount consumed and their bioavailability [55]. However, that does not mean that higher dietary intakes are absolutely exploited by human metabolism [56]. Nonetheless, it is well documented that adherence to a healthy dietary pattern rich in polyphenols, like the Mediterranean diet, is directly and inversely related to chronic disease risk development, including BC [14,57]. The generation of reactive oxygen species (ROS) is a feature of mitochondrial dysfunction, and elevated levels of these species have been reported as supporters of factors that can drive tumor initiation, progression, and metastasis [58,59]. Several dietary antioxidants may protect cells from oxidative stress by scavenging free radicals and quenching lipid peroxidation chain reactions, both of which have a significant potential for DNA damage and genomic instability [60].

Many natural products and their derivatives are under research as agents for cancer prevention and treatment due to their enormous structural and chemical variety [61,62]. Natural products are modulators of a wide range of enzymes and cell receptors, critical for signaling pathways [32,56,63]. A growing emphasis is being placed on alternative medicine and dietary approaches toward the prevention and treatment of BC [64,65]. There is an immediate need for more efficient, with fewer side effects, therapeutic, and preventive strategies, and screening natural products from plants increases the chances of achieving this goal [66]. There are many classes of plant-derived cytotoxic natural products studied for further improvement and development of drugs, such as phenolic acids, flavonoids, tannins, coumarins, lignans, lignins, naphtoquinones, anthraquinones, xanthones, and stilbenes, which are well-known phenolic compounds found in plant *taxa* [67,68].

For instance, the presence of the polyphenol Rosmarinic Acid in medicinal plants, herbs, and spices has beneficial and health-promoting effects [69]. Rosmarinic Acid is one of the metabolites that is actually produced on a large scale by in vitro cultures [70]. Rosmarinic Acid in plants is an indicator of growth and defense. It acts repressively in many cancer types by interfering with the signaling pathways related to the upregulation of metastasis [63]. Preventing metastasis can be achieved by arresting any of the three stages of cancer development: initiation, promotion, and progression [71].

### 2.1. History of Rosmarinic Acid

Rosmarinic Acid (C_18_H_16_O_8_) is an ester of caffeic acid and 3-(3,4-dihydroxyphenyl)lactic acid. Rosmarinic Acid is formally known as (R)-α-[[3-(3,4-dihydroxyphenyl)-1-oxo-2E-propenyl]oxy]-3,4-dihydroxy-enzenepropanoic acid and has a chiral center with S(-) and R(+) enantiomers (Figure 1) [70]. Rosmarinic Acid was first isolated in 1958 as a pure compound by two Italian chemists, Scarpati and Oriente [72]. The compound was named after the rosemary plant (*Rosmarinus officinalis* L.) it was extracted from and was elucidated as an ester of 3,4-dihydroxy-phenyllactic acid [70].

However, Rosmarinic Acid and many of its derivatives were already known as “Labiatengerbstoffe”, tannin-like compounds found in species of the *Lamiaceae* family [72]. Before Rosmarinic Acid’s structure was elucidated, it was assumed that the tannin-like compound from *Melissa officinalis* contained caffeic acid and could not be a gallotannin, ellagitannin, or condensed catechin [72]. Eventually, biogenetic studies in *Mentha* plants [73] and cell line experiments revealed that phenylalanine and tyrosine, two aromatic amino acids, were the precursors of Rosmarinic Acid [74]. The chemical synthesis of Rosmarinic Acid was completed in 1991 by Albrecht, and ever since, many derivatives and stereoisomers of it have been synthesized [72].

### 2.2. Natural Occurrences of Rosmarinic Acid

Rosmarinic Acid is distributed among thirty-nine plant families [70,72]. More specifically, it is found in numerous species of the *Boraginaceae* family and the *Nepetoideae* subfamily of the *Lamiaceae taxa* (Table 1) [50,75,76]. Rosmarinic Acid is the main compound in the extracts of *Basilicum polystachyon*, but it is reported in many other plants, including *Thymus mastichina* and plants belonging to the *Agastache* genus of the *Lamiaceae* family [77,78]. Rosmarinic acid is also present in several species of the *Labiatae* family (Table 2). It exists in vacuoles and the cytoplasm (as an anion) of *Mentha spicata*, and its diffusion between membranes is impossible [70]. Additionally, this phenolic acid is accumulated in the *Choranthaceae* and *Blechnaceae taxa* and several marine hydrophilus angiosperms, e.g., *Zostera marina Linnaeus* [50]. Rosmarinic Acid is present either in simple or evolving land plants, as well as in *monocotyledonous* and *eudicotyledonous* species [50]. Rosmarinic Acid is widely distributed in the plant kingdom, but it can be considered an important chemotaxonomic marker for the *Lamiaceae* family only [70,77].

With regard to Rosmarinic Acid content in plants, a comparative study showed that this phenolic compound reaches up to 58.5 mg/g of the dried plant. Among the species with the highest Rosmarinic Acid content are the *Mentha* species, more specifically, *M. spicata* [75].

### 2.3. Biosynthesis of Rosmarinic Acid

Except for *Boraginaceae* and *Lamiaceae*, many kinds of ferns and seagrass are the “natural factories” where the biosynthesis of Rosmarinic Acid happens, with the contribution of eight enzymes usually present in the biosynthesis pathways of flavonoids [70,72]. The precursors of Rosmarinic Acid are the aromatic amino acids L-phenylalanine and L-tyrosine [70,72]. The first one is ultimately converted into caffeic acid, while the second is transformed into 3,4-dihydroxyphenyllactic acid through several stages [73].

Initially, L-phenylalanine is transformed into t-cinnamic acid by phenylalanine ammonia-lyase (PAL). Cytochrome P450 monooxygenase cinnamate 4-hydroxylase (C4H) leads to the formation of 4-coumaric acid by hydroxylation of t-cinnamic acid in position 4. The product of the next reaction, 4-coumaroyl-CoA, is created due to the action of coenzyme A ligase (4CL) on 4-coumaric acid. 4-coumaroyl-CoA is an activated molecule that was detected in suspension cells of *Coleus blumei* (and the previous three products, too) [72,79]. 4-coumaroyl-CoA acts as a donor of hydroxycinnamate in a compound that is formed from tyrosine.

In the first reaction of L-tyrosine, 2-oxoglutarate is used as a co-substrate, and the main product is a transaminated form of the amino acid 4-hydroxyphenylpyruvic acid (pHPP) and glutamate. The enzyme that takes part in it is the pyridoxalphosphate-dependent transaminase tyrosine aminotransferase (TAT). This stage is a crossroad between the biosynthesis of Rosmarinic Acid, tocopherols, and plastoquinones due to the action of hydroxyphenylpyruvate dioxygenase (HPPD) on 4-hydroxyphenylpyruvate, which leads to homogentisic acid, the precursor of tocopherols and plastoquinones. For Rosmarinic Acid biosynthesis, the substrate 4-hydroxyphenylpyruvate is transformed by NAD(P)H-dependent hydroxyphenylpyruvate reductase (HPPR) into 4-hydroxyphenyllactic acid (pHPL), more specifically, the R(+)-stereoisomer of hydroxyphenyllactate. 3,4-dihydroxyphenylpyruvate is reduced by HPPR but with low affinity [70]. However, 3,4-dihydroxyphenylalanine (DOPA) may also provide a 4-hydroxyphenyllactate moiety [70].

These two compounds, 4-coumaroyl-CoA and pHPL, are the final substrates for the formation of Rosmarinic Acid, with the contribution of the enzyme synthase of rosmarinic acid (RAS). The ester 4-coumaroyl-4′-hydroxyphenyllactic acid (4C-pHPL), after the action of two cytochrome P450 monooxygenases that catalyze its hydroxyliation, is transformed into Rosmarinic Acid (Figure 2). All enzymes involved in the biosynthesis of Rosmarinic Acid have been extensively studied, and information about the relative genes is already known.

Albrecht developed the chemical synthesis of Rosmarinic Acid, which had long been sought for [72]. Since then, a variety of chemical synthesis methods for Rosmarinic Acid and its derivatives, such as the methyl ester, various stereoisomers, and the less hydroxylated isorinic acid, have been published [72]. Additionally, Rosmarinic Acid can be synthesized from vertraldethyde through the Erlenmeyer reaction, hydrolytic loop-open reduction, carboxyl protection, esterification, the Kunz–Waldmann procedure, and demethylation [80]. Very recently, a high overall yield and complete synthesis of Rosmarinic Acid and similar ester derivatives from (+)-danshensu (or comparable aryl lactic acid) and related hydroxybenzaldehydes have been presented [81].

### 2.4. Effects of Rosmarinic Acid in MDA-MB-231 and MDA-MB-468 TNBC Cells

A plethora of pharmacological properties has been attributed to Rosmarinic Acid, including antiviral, antibacterial, astringent, antimutagenic, anti-inflammatory, antiallergic, and antioxidant activities, with the latter two being the most widely known and extensively studied [50,63,70,72,77,82,83]. Studies have shown that Rosmarinic Acid also acts as an immunomodulatory and neuroprotective factor [83]. Rosmarinic Acid’s contribution to the reduction in tumor development has been noticed in many cancer types, including colon, breast, liver, stomach, and lung, as well as melanoma and leukemia [77,84].

There is strong evidence that Rosmarinic Acid could be a potential therapeutic agent for several types of BC because it modulates several signaling pathways that lead to tumor tissue development [70,71,85]. It should be noted that all studies presented in this review showed that the intensity of the apoptotic effect induced by Rosmarinic Acid is dose-dependent [71,83,86].

The estrogen receptor and progesterone receptor are proteins directly related to BC, and, in combination with HER-2/neu gene amplification, they are used for the classification of TNBC, a very aggressive and metastatic subtype that lacks the above three significant molecular markers [85,87]. HER2/neu is a proto-oncogene located on the long arm of chromosome 17, and the presence of HER2/neu gene amplification is prognostically and therapeutically significant for patients with breast cancer [87]. The amplification of the HER-2/neu oncogene is associated with shorter disease-free survival in 25% of breast tumors [88]. This cancer subtype is unaffected by hormone therapy, and the expectations of TNBC treatment (only with chemotherapy) are rather limited [27].

MCF-7 (invasive breast ductal carcinoma) and T-47D (human breast cancer cells) are human hormone-dependent BC cell lines (ER/PR-positive), which, together with human breast adenocarcinoma estrogen-independent cells (i.e., MDA-MB-231 and MDA-MB-468), have been studied extensively. Rosmarinic Acid acts cytotoxically and antiproliferatively in these two TNBC cell lines (Figure 3) and can be a potential approach to treatment [71,83]. Rosmarinic Acid seems to cause cell cycle arrest and apoptosis in distinct ways. In fact, Rosmarinic Acid arrests the G0/G1 phase in MDA-MB-231 cells and the S-phase in MDA-MB-468 cells following apoptosis (interruption of the G2/M process). Rosmarinic Acid, however, has been shown to halt the cell cycle of MDA-MB-468 cells early in mitosis, having a greater apoptotic impact compared to MDA-MB-231 cells [83]. In the same study, Rosmarinic Acid also affected the expression of the genes relevant to the apoptosis event and was more noticeable in MDA-MB-468 cells. More specifically, Rosmarinic Acid enhanced the expression of TNF (tumor necrosis factor), GADD45A (growth arrest and DNA damage-inducible 45 alpha), and the proapoptotic BNIP3 (Bcl-2 interacting protein 3, a biomarker for the identification of tumors) genes [83]. On the other hand, MDA-MB-231 cells appeared less sensitive to Rosmarinic Acid, and the reduction in cell viability was attributed to changes in the expression of the following genes: upregulation of BNIP3, TNFRSF25 (tumor necrosis factor receptor superfamily 25), and HRK (harakiri) [83]. Furthermore, Rosmarinic Acid induced the suppression of three genes: ligand TNFSF10 (TNF superfamily member 10) and BIRC5 (baculoviral IAP repeat-containing 5, a prognostic biomarker associated with tumor immune cell infiltration) in MDA-MB-468 cells, and TNFRSF11B (TNF receptor superfamily 11B) in MDA-MB-231 cells [83].

Apoptotic signaling can be initiated in the nucleus, mitochondria, or cell membrane and proceed by several routes [83,89]. BIRC5 (Survivin) inhibition appears to be the most important effect of Rosmarinic Acid on MDA-MB-468 cells in contrast to MDA-MB-231 cells because this protein, except for its prognostic significance in cancer diagnosis, is largely responsible for the reduced performance of chemotherapy and radiotherapy. It is an important member of the inhibitor of apoptosis (IAP) family, and it is expressed in the G2-M phase, which is selectively interrupted in MDA-MB-468 cells by Rosmarinic Acid [90]. Therefore, the control of Survivin by Rosmarinic Acid might provide a new target for cancer therapy.

Li et al. showed that in MDA-MB-231 cells, the Bcl-2 gene is downregulated while the Bax gene expression is increased in the presence of Rosmarinic Acid [86]. BAX expression is controlled by the tumor suppressor protein p53 and has been linked to p53-mediated apoptosis [91]. The p53 protein is a transcription factor that controls numerous downstream target genes, including BAX, when activated as part of the cell’s response to stress [91]. The Bcl-2 protein increases cell death resistance in aggressive metastatic phenotypes with out-of-control expression. Therefore, the association and the ratio of BAX to Bcl-2 are determining indicators of the survival or death of a cell following an apoptotic stimulus [92]. The same research group also investigated the effect of Rosmarinic Acid on breast cancer stem-like cells (BCSCs) derived from MDA-MB-231 cells [41,93]. BCSCs’ response to Rosmarinic Acid leads to a substantial decrease in proliferation and migration through the Bcl-2/Bax signaling pathways. Furthermore, the authors revealed the involvement of the Hedgehog (Hh) pathway in BC development. The Hh pathway modulates cell growth and survival, while inhibiting its key regulators can block the development or metastasis of cancer cells with irreversible cell death [93]. The experiments showed that Rosmarinic Acid inhibited Hh signaling genes’ expression in BCSCs.

Another study found that rosemary extract with Rosmarinic Acid and carnosic acid as primary ingredients inhibited cancer cell viability in the ER+, HER2+, and TNBC subtypes (MDA-MB-231 and MDA-MB-468 cells) [71]. MDA-MB-231 cells are extremely invasive; hence, preventing metastasis is critical. It is worth noting that the nuclear factor kappa-light-chain-enhancer of activated B cells (NF-κB) signaling pathway, which is frequent in TNBC cell lines, may be beneficial in the development of anticancer drugs for TNBCs [94,95]. Stress, cytokines, free radicals, ultraviolet irradiation, and oxidized low-density lipoproteins (oxLDLs) are all examples of stimuli that activate NF-κB. Cancer, inflammatory and autoimmune disorders, and abnormal immunological development have all been associated with NF-κB [94,96,97,98]. The inhibition of NF-κB increases chemotherapy and radiation results and hormone response, which is associated with increased disease-free survival in BC patients [94].

The NF-κB pathway involves a series of overlapping reactions that lead to the expression of a group of cytokines, including interleukin-8 (IL-8), to promote colony formation. The production of IL-8 happens when the receptor activator of the NF-κB ligand (RANKL) binds to its receptor RANK. The activation is prevented by the presence of osteoprotegerin (OPG), a tumor necrosis factor (TNF) of the cytokine family produced by osteoblasts [99]. While OPG binds with RANKL due to structural similarity with RANK, NF-κB remains inactivated, and the production of IL-8 stops at this significant stage.

IL-8 is an inflammatory factor and an important biomarker of disease prognosis. The NF-κB pathway is regulated by Rosmarinic Acid, which, in turn, seems to inhibit bone metastasis from breast carcinoma by preventing the expression of IL-8. Although the exact nature of the interaction remains unclear, the antimetastatic effect of Rosmarinic Acid may be attributed to its antioxidant properties. To this point, it has been suggested that Rosmarinic Acid inhibits NF-κB signaling via ROS scavenging or suppression or by stimulating antioxidant enzymes.

MARK4 (Microtubule affinity-regulating kinase 4) is a key regulator of many signaling pathways (such as NF-κB), and it is expressed in almost every organ of the human body [100]. It is involved in microtubule organization in neuronal cells, and its aberrant expression is responsible for obesity, diabetes mellitus, neurodegenerative diseases, and metastatic breast carcinomas. MARK4 controls numerous cell activities, including microtubule dynamics, cell division, cell cycle regulation, and apoptosis [63,100]. The dysregulation of MARK4 causes cell death; therefore, the investigation of natural products and their molecular assembly into a target protein is a promising approach for anticancer drug design [100]. The interaction between Rosmarinic Acid and the MARK4 protein fits perfectly in the above philosophy: molecular docking analysis and molecular dynamics (MDs) simulation have shown Rosmarinic Acid’s high binding affinity to the MARK4 enzyme, attributed to a great number of hydrogen bonds to its catalytic domain [63]. In vitro experiments in MDA-MB-231 cancer cells treated with Rosmarinic Acid have shown that proliferation and migration were significantly attenuated, and eventually, cells were led to apoptosis [63]. The above can be explained by the fact that MARK4’s overexpression is associated with tumor development, and Rosmarinic Acid forms an extra-stable complex, hampering its activity [63].

Due to MARK4’s loss of function, Hippo signaling is also intercepted: the transcription factors YAP and TAZ, essential transducers of the cell’s structural features and solid tumor developers, remain inactivated, and YAP/TAZ target gene expression decreases [10]. When activated, these transcription factors induce cancer stem cell attributes, chemoresistance, and metastasis [10]. Through YAP/TAZ inhibition, the upregulation of Bcl2-family members is prevented, and the mitochondrial-induced apoptosis pathway is enhanced, induced by the activity of TNF-α (tumor necrosis factor-alpha) and FAS ligands [101]. The TAZ protein’s expression is a major issue, especially for the TNBC subclass. In vitro experiments in MDA-MB-231 cells showed that MARK4 depletion attenuated the phosphorylation of the core components of the Hippo pathway and caused the translocation of nucleus YAP/TAZ to the cytoplasm [101]. As a result, weakening nucleus YAP/TAZ targets decreased their expression. Additionally, the absence of MARK4 activity had a negative effect on the Hippo pathway, suppressing the tumorigenic properties of MDA-MB-231 cancer cells [10,102].

Until now, there have been very few data regarding Rosmarinic Acid’s side effects and toxicity for humans. On the contrary, there is evidence that the antioxidant activity of Rosmarinic Acid can limit the toxicity exerted by chemotherapy in human HL-60 promyelocytic leukemia cells [103]. Additionally, in vivo studies have shown that Rosmarinic Acid decreased the hepatic and renal toxicity induced by methotrexate, as well as the cardiotoxicity of doxorubicin [39,104,105]. In a randomized, placebo-controlled trial in BC patients, oral intake of *Prunella vulgaris* L., rich in Rosmarinic Acid, inhibited chemotherapy (taxanes) side effects, namely neutrophil-reduced fever and anemia [106].

## 3. Conclusions

In conclusion, the overexpression of proapoptotic proteins and oncogenic stress factors, as well as the repression of tumor-associated genes, is a promising way to prevent the metastasis of TNBC. In vitro studies suggest that Rosmarinic Acid interferes with several interconnected processes, including cell proliferation, angiogenesis, cell adhesion, migration, and invasion into the surrounding tissue, leading to enhanced apoptosis in a dose-dependent manner. Nevertheless, in vivo studies are essential to confirm in vitro results and assess the safety and efficacy of Rosmarinic Acid as an antitumor agent. Additionally, a thorough toxicity examination of Rosmarinic Acid, including acute toxicity, chronic toxicity, LD_50_, and therapeutic window, should be carried out in the near future.

## Figures and Tables

**Figure 1 nutrients-16-00002-f001:**
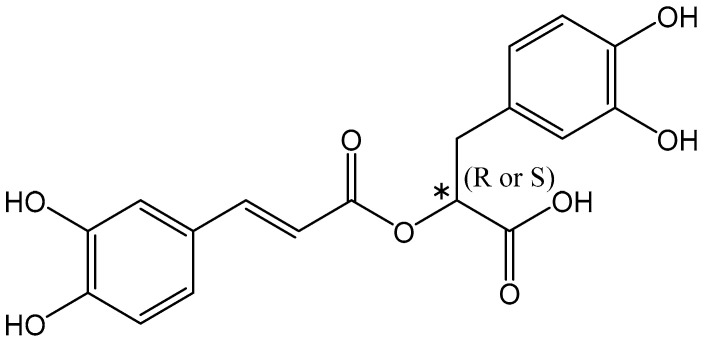
Chemical structure of Rosmarinic Acid. * Ιndicates the presence of a stereogenic center.

**Figure 2 nutrients-16-00002-f002:**
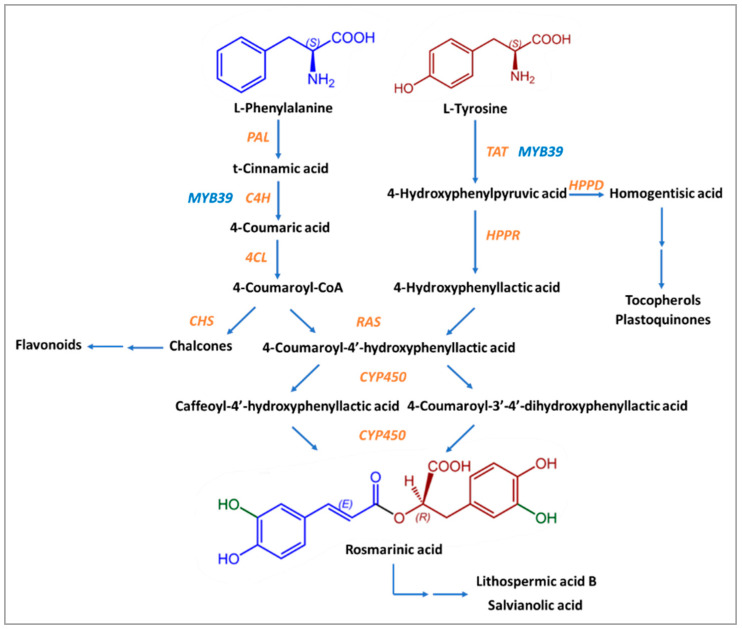
Proposed biosynthetic pathway of Rosmarinic Acid. Enzymes are shown with in orange and blue letters. Enzyme substrates are shown in black letters. The chemical structure derived from L-Phenylalanine is shown in blue and the chemical structure derived from L-Tyrosine is shown in brown-red. Green color shows the hydroxyl groups that come from the action of enzyme CYP450.

**Figure 3 nutrients-16-00002-f003:**
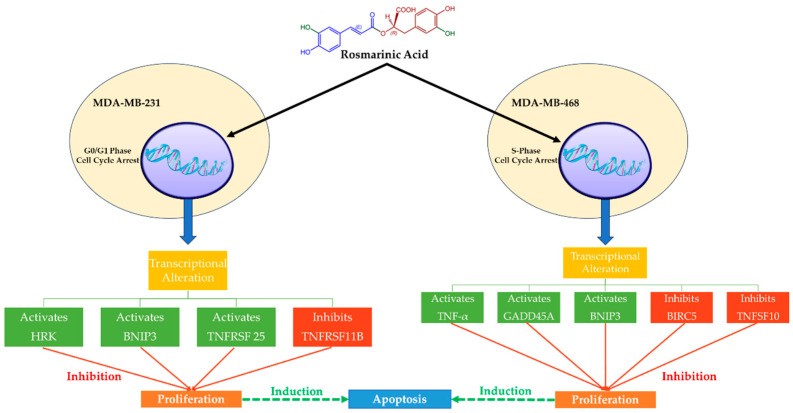
The mechanisms of action of Rosmarinic Acid on MDA-MB-231 and MDA-MB-468 TNBC cell lines. Rosmarinic Acid is cytotoxic and antiproliferative in two TNBC cell lines, MDA-MB-231 and MDA-MB-468, which suggests that it might be used as a therapeutic. Following apoptosis, Rosmarinic Acid regulates the cell cycle, halting the G0/G1 phase in MDA-MB-231 cells and the S-phase in MDA-MB-468 cells. Rosmarinic Acid increases the expression of TNF, GADD45A, and BNIP3 genes in MDA-MB-468 cells but not in MDA-MB-231 cells. Rosmarinic Acid causes the repression of three genes, one of which is BIRC5 (Survivin), which is responsible for decreased chemotherapy and radiation efficacy. Survivin control by Rosmarinic Acid might offer a novel target for cancer treatment.

**Table 1 nutrients-16-00002-t001:** Natural occurrences of Rosmarinic Acid in species of *Lamiaceae* and *Boraginaceae* families.

*Lamiaceae taxa*			
*Ajuga*	*Agastache*	*Calamintha*	*Cedronella*
*Coleus*	*Collimsonia*	*Dracocephalum*	*Elsholtzia*
*Glechoma*	*Hornium*	*Lavandula*	*Lycopus*
*Melissa*	*Mentha*	*Micromeria*	*Monarda*
*Origanum*	*Perilla*	*Perovskia*	*Plectranthus*
*Salvia*	*Satureja*	*Thymus*	
*Boraginaceae taxa*			
*Cerinthe*	*Echium*	*Heliotropium*	*Lindefolia*
*Lithospermum*	*Nonea*	*Symphytum*	*Hydrophyllum*
*Nemophila*	*Phacelia*		

Table information is drawn from [50,75,76].

**Table 2 nutrients-16-00002-t002:** Natural occurrences of Rosmarinic Acid in species of *Labiatae* family.

*Salvia officinalis*	*Rosmarinus officinalis*	*Mentha piperita*
*Salvia limbata*	*Lavandula angustifolia*	*Mentha pulegium*
*Salvia virgata*	*Thymus daenensis*	*Mentha longifolia*
*Salvia hypoleuca*	*Thymus citriodorous*	*Mentha spicata*
*Salvia macrosiphon*	*Thymus pubescens*	*Mentha aquatica*
*Salvia choloroleuca*	*Thymus vulgaris*	*Mentha crispa*
*Melissa officinalis*	*Zataria multiflora*	*Zhumeria majdae*
*Origanum vulgare*	*Ocimum sanctum*	*Perovskia artemisoides*
*Satureja khuzistanica*	*Satureja atropatana*	*Satureja macrantha*
*Satureja bachtiarica*	*Satureja mutica*	*Satureja hortensis*
*Forsythia koreana*	*Hyptis pectinata*	

Table information is drawn from [77,78].

## Data Availability

All data and analysis are available within the manuscript.

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
