# Peer review of "Molecular Pathways of Rosmarinic Acid Anticancer Activity in Triple-Negative Breast Cancer Cells: A Literature Review"

_nutrients, 2023, doi:10.3390/nu16010002_

Round 1

Reviewer 1 Report

Comments and Suggestions for Authors

The natural products derived from plants provide a foundation for the exploration of natural medicines. Rosmarinic acid is one of the important plant active natural products. Therefore, a review of the activity and function of rosmarinic acid is of great significance. However, the depth of this review is still slightly lacking, and it is recommended to add contents to provide more foundation for related researcher. There are several suggestions as follow.

1. The title focuses too much on breast cancer, but it does not cover the contents of this manuscript.

2. It is not recommended to use abbreviations for the substance of a word. For example, it is not recommended to use PP as the abbreviation for polyphenols.

3. The concentrations of rosmarinic acid are crucial for subsequent extraction and obtaining more amounts. It is recommended to increase the data related to the concentrations of rosmarinic acid in plants. In addition, chemical synthesis is also an important means to solve the content of natural products, so it is necessary to summarize the pathways for the chemical synthesis of rosmarinic acid.

4. The information in Figure 3 is not displayed clearly, please modify.

5. Pay attention to the format of the entire text, especially the format of the references is not uniform.

Author Response

All authors would like to express their gratitude to the reviewers for the positive feedback and constructive comments on the manuscript nutrients-2772859. We believe that all suggestions have resulted in an improved version, which you will find uploaded. The manuscript has been revised to address reviewers’ comments (given alongside our responses) and the journal’s instructions to reduce manuscript duplicate rate. All authors agree with this.

Reviewer 1

The natural products derived from plants provide a foundation for the exploration of natural medicines. Rosmarinic acid is one of the important plant active natural products. Therefore, a review of the activity and function of rosmarinic acid is of great significance. However, the depth of this review is still slightly lacking, and it is recommended to add contents to provide more foundation for related researcher. There are several suggestions as follow.

Thank you for your comment. According to your suggestions we revised our manuscript providing more content regarding the (a) rosmarinic content in plants, (b) rosmarinic acid chemical synthesis and (c) the effects of rosmarinic acid on TNBC.

  1. The title focuses too much on breast cancer, but it does not cover the contents of this manuscript.

In the present review we discuss the in vitro anti-cancer activity of rosmarinic acid in Triple Negative Breast Cancer (MDA-MB-231 and MDA-MB-468 cells) and the implicated molecular pathways. Therefore, according to your suggestion we revised the title to: “Molecular Pathways of Rosmarinic Acid Anti-cancer Activity in Triple Negative Breast Cancer Cells: A Literature Review”.

  1. It is not recommended to use abbreviations for the substance of a word. For example, it is not recommended to use PP as the abbreviation for polyphenols.

Thank you for your comment. We revised the whole manuscript and removed the abbreviations used for polyphenols, natural plants and rosmarinic acid as well.

  1. The concentrations of rosmarinic acid are crucial for subsequent extraction and obtaining more amounts. It is recommended to increase the data related to the concentrations of rosmarinic acid in plants. In addition, chemical synthesis is also an important means to solve the content of natural products, so it is necessary to summarize the pathways for the chemical synthesis of rosmarinic acid.

Thank you for your comment. We added information regarding rosmarinic acid content in plants (lines 197-200) and rosmarinic acid chemical synthesis (lines 241-250).

  1. The information in Figure 3 is not displayed clearly, please modify.

Thank you for your comment. Following your instructions, we modified the information in the caption of Figure 3 with brief explanation of rosmarinic acid bioactivity in TNBC cell lines.

  1. Pay attention to the format of the entire text, especially the format of the references is not uniform.

Following your instructions, we fixed text and bibliography formatting according to Journal’s instructions.

Reviewer 2 Report

Comments and Suggestions for Authors

Dear Editor,

Enclosed please find the comments on the following manuscript:

Journal: Nutrients

Manuscript ID: nutrients-2772859

Type of manuscript:

Title: Effects of Rosmarinic Acid in Triple Negative Breast Cancer Cells: A Literature Review

The manuscript “Effects of Rosmarinic Acid in Triple Negative Breast Cancer Cells: A Literature Review” by Evangelia K. Konstantinou et al., is a great phytochemical review in a current and interesting topic and could be of interest for natural product and pharmaceutical scientists. Authors collected the in vitro, in vivo and clinical evidences to treatment breast cancer cells. This review may contribute to pharmaceutical application and medical research. In my pion, this review is acceptable after minor revision.

Minor comments:

Page 4, Authors should correct the typing error.

Table 2. Natural occurrences of RA in Mentha spicata?

Page 9, Authors should revise following paragraph: Moreover, in patients with BC, treatment with RA inhibited side effects, namely, neutrophil-reduced fever and anemia caused by chemotherapy [106].

This clinical trial uses the traditional Chinese medicine (Prunella vulgaris) to reduce the side effects of taxanes.

Page 13, Author should correct the scientific name of references [106].  

The scientific name is always italicized.

Author Response

All authors would like to express their gratitude to the reviewers for the positive feedback and constructive comments on the manuscript nutrients-2772859. We believe that all suggestions have resulted in an improved version, which you will find uploaded. The manuscript has been revised to address reviewers’ comments (given alongside our responses) and the journal’s instructions to reduce manuscript duplicate rate. All authors agree with this.

Reviewer 2

The manuscript “Effects of Rosmarinic Acid in Triple Negative Breast Cancer Cells: A Literature Review” by Evangelia K. Konstantinou et al., is a great phytochemical review in a current and interesting topic and could be of interest for natural product and pharmaceutical scientists. Authors collected the in vitro, in vivo and clinical evidences to treatment breast cancer cells. This review may contribute to pharmaceutical application and medical research. In my pion, this review is acceptable after minor revision. Minor comments:

Page 4, Authors should correct the typing error.

Thank you for your comment. Following your instructions, we fixed all typing errors in the manuscript.

Table 2. Natural occurrences of RA in Mentha spicata?

We corrected the title of the table to “Natural occurrences of Rosmarinic Acid in 29 species of Labiatae family”. Thank you. 

Page 9, Authors should revise following paragraph: Moreover, in patients with BC, treatment with RA inhibited side effects, namely, neutrophil-reduced fever and anemia caused by chemotherapy [106]. This clinical trial uses the traditional Chinese medicine (Prunella vulgaris) to reduce the side effects of taxanes.

Thank you for your comment. We revised the sentence to: In a randomized, placebo-controlled trial in BC patients, oral intake of Prunella vulgaris L., rich in RA, inhibited chemotherapy (taxanes) side effects, namely neutrophil-reduced fever and anemia [107].

Page 13, Author should correct the scientific name of references [106].  The scientific name is always italicized.

Thank you for your comment. We have revised the whole manuscript according to your suggestion and italicized scientific names of plants.